# DNA Damage Response in Early Breast Cancer: A Phase III Cohort in the Phobos Study

**DOI:** 10.3390/cancers16152628

**Published:** 2024-07-23

**Authors:** Eriseld Krasniqi, Cristiana Ercolani, Anna Di Benedetto, Francesca Sofia Di Lisa, Lorena Filomeno, Teresa Arcuri, Claudio Botti, Fabio Pelle, Flavia Cavicchi, Sonia Cappelli, Maddalena Barba, Laura Pizzuti, Marcello Maugeri-Saccà, Luca Moscetti, Antonino Grassadonia, Nicola Tinari, Giuseppe Sanguineti, Silvia Takanen, Davide Fragnito, Irene Terrenato, Simonetta Buglioni, Letizia Perracchio, Agnese Latorre, Ruggero De Maria, Matteo Pallocca, Gennaro Ciliberto, Francesco Giotta, Patrizia Vici

**Affiliations:** 1Phase IV Clinical Studies Unit, IRCCS Regina Elena National Cancer Institute, 00144 Rome, Italy; eriseld.krasniqi@ifo.it (E.K.); francescasofia.dilisa@ifo.it (F.S.D.L.); teresa.arcuri@ifo.it (T.A.); patrizia.vici@ifo.it (P.V.); 2Pathology Department, IRCCS Regina Elena National Cancer Institute, 00144 Rome, Italy; cristiana.ercolani@ifo.it (C.E.); anna.dibenedetto@ifo.it (A.D.B.); simonetta.buglioni@ifo.it (S.B.); letizia.perracchio@ifo.it (L.P.); 3Department of Radiological, Oncological and Anatomo-Pathological Sciences, Sapienza University of Rome, 00185 Rome, Italy; 4Breast Surgery Department, IRCCS Regina Elena National Cancer Institute, 00144 Rome, Italy; claudio.botti@ifo.it (C.B.); fabio.pelle@ifo.it (F.P.); flavia.cavicchi@ifo.it (F.C.); sonia.cappelli@ifo.it (S.C.); 5Division of Medical Oncology 1, IRCCS Regina Elena National Cancer Institute, 00144 Rome, Italy; maddalena.barba@gmail.com (M.B.); laura.pizzuti@ifo.it (L.P.); 6Clinical Trial Center, Biostatistics and Bioinformatics Unit, Department of Research, Diagnosis and Innovative Technologies, IRCCS Regina Elena National Cancer Institute, 00144 Rome, Italy; marcello.maugerisacca@ifo.it (M.M.-S.); irene.terrenato@ifo.it (I.T.); 7Division of Medical Oncology 2, IRCCS Regina Elena National Cancer Institute, 00144 Rome, Italy; 8Oncology and Hemathology Department, Azienda Ospedaliero-Universitaria Policlinico di Modena, 41125 Modena, Italy; moscetti.luca@aou.mo.it; 9Department of Innovative Technologies in Medicine and Dentistry, Center for Advanced Studies and Technology (CAST), G. D’Annunzio University Chieti-Pescara, 66100 Chieti, Italy; grassa@unich.it; 10Department of Medical, Oral and Biotechnological Sciences, Center for Advanced Studies and Technology (CAST), G. D’Annunzio University Chieti-Pescara, 66100 Chieti, Italy; ntinari@unich.it; 11Department of Radiation Oncology, IRCCS Regina Elena National Cancer Institute, 00144 Rome, Italy; giuseppe.sanguineti@ifo.it (G.S.); silvia.takanen@ifo.it (S.T.); 12Institute of Endocrinology and Experimental Oncology “G Salvatore”, National Research Council (CNR), 00186 Naples, Italy; m.pallocca@ieos.cnr.it; 13Medical Oncology, IRCCS Istituto Tumori “Giovanni Paolo II”, 70124 Bari, Italy; a.latorre@oncologico.bari.it (A.L.); f.giotta@oncologico.bari.it (F.G.); 14Department of Translational Medicine and Surgery, Catholic University of the Sacred Hearth, 00153 Rome, Italy; demariaruggero@gmail.com; 15IRCCS Fondazione Policlinico Universitario “A Gemelli”, 00168 Rome, Italy; 16Scientific Direction, IRCCS Regina Elena National Cancer Institute, 00144 Rome, Italy; gennaro.ciliberto@ifo.it

**Keywords:** breast cancer, DNA damage response and repair, adjuvant chemotherapy, prognostic biomarkers, predictive factors

## Abstract

**Simple Summary:**

This study evaluates DDR biomarkers in 222 node-positive early breast cancer patients from a Phase III trial on adjuvant taxanes. Over a 234-month follow-up period, no differences in DFS or OS were observed between treatment groups. However, a DDR risk score, influenced by ATM and ATR expression, proved to be an independent prognostic factor for both DFS and OS. Validation in a public cohort confirmed ATM’s protective role. These findings highlight the importance of DDR pathways in breast cancer prognostication and support, integrating molecular markers with clinical–pathological factors to inform treatment strategies.

**Abstract:**

We assessed the impact of DNA damage response and repair (DDR) biomarker expressions in 222 node-positive early breast cancer (BC) patients from a previous Phase III GOIM 9902 trial of adjuvant taxanes. At a median follow-up of 64 months, the original study showed no disease-free survival (DFS) or overall survival (OS) differences with the addition of docetaxel (D) to epirubicine-cyclophosphamide (EC). Immunohistochemistry was employed to assess the expression of DDR phosphoproteins (pATM, pATR, pCHK1, γH2AX, pRPA32, and pWEE1) in tumor tissue, and their association with clinical outcomes was evaluated through the Cox elastic net model. Over an extended follow-up of 234 months, we confirmed no significant differences in DFS or OS between patients treated with EC and those receiving D → EC. A DDR risk score, inversely driven by ATM and ATR expression, emerged as an independent prognostic factor for both DFS (HR = 0.41, *p* < 0.0001) and OS (HR = 0.61, *p* = 0.046). Further validation in a public adjuvant BC cohort was possible only for ATM, confirming its protective role. Overall, our findings confirm the potential role of the DDR pathway in BC prognostication and in shaping treatment strategies advocating for an integrated approach, combining molecular markers with clinical–pathological factors.

## 1. Introduction

Breast cancer (BC), the second most common type of cancer globally, represents a major health issue and is one of the leading causes of death in women [1,2]. Despite this, advances in early detection and treatment have significantly improved outcomes. Currently, women diagnosed with early/locally advanced BC and who have access to adequate treatment face approximately a 90% chance of cure, with a wide range of treatment options available [3]. Adjuvant systemic treatment post-radical surgery, aims to minimize recurrence risk and enhance survival, employing chemotherapy, endocrine therapy and targeted agents [4,5]. Treatment choice is complex and tailored, considering both the biological characteristics of the cancer and patient-related factors [6]. Recent advances have enabled more personalized treatment approaches, significantly impacting BC mortality and long-term quality of life. The primary factors guiding systemic adjuvant treatment decisions are disease extent and classification based on intrinsic BC subtypes: luminal A, luminal B (human epidermal growth factor receptor 2 (HER2)-negative and HER2-positive), HER2-enriched, and basal-like (triple-negative) subtypes [7]. The standard of care for high-risk post-surgical BC patients includes adjuvant chemotherapy with an anthracycline-containing regimen followed or preceded by a taxane, with the addition of anti-HER2 treatment in HER2-positive disease [8,9,10]. Despite these advances, limitations in current treatments remain. These limitations include variability in treatment response and the potential for significant side effects, which underscore the need for more precise biomarkers to guide therapy decisions.

Central to tumor prognosis and chemotherapy efficacy is the DNA Damage Response and Repair (DDR) pathway. Despite the critical role of DDR in cellular recovery from DNA damage, research shows conflicting results regarding the prognostic and predictive value of specific genes and proteins within these pathways [11,12,13,14]. There are two kinases at the core of the DDR, ATM (ataxia telangiectasia mutated) and ATR (ataxia telangiectasia and Rad3 related), which are particularly activated in response to single and double-strand DNA breaks, orchestrating a complex mechanism that encompasses the entire spectrum from damage detection to repair initiation [15,16]. Key to this process is the involvement of auxiliary components such as replication protein A 32 (RPA32) and H2A Histone Family Member X (H2AX), which are both crucial in the early detection phase of DNA strand breaks [17,18]. Furthermore, the role of Checkpoint Kinase 1 (CHK1) and WEE1 G2 Checkpoint Kinase (WEE1) in cell cycle arrest and in maintaining the integrity of the response mechanism is indispensable [19,20]. 

This study aims to evaluate the expression of active DDR proteins and their impact on disease course and clinical outcomes in the adjuvant setting of breast cancer patients. By providing a detailed analysis of DDR marker expression, this research seeks to offer new insights into their potential as prognostic and predictive biomarkers, ultimately contributing to more personalized and effective treatment strategies for breast cancer.

## 2. Materials and Methods

### 2.1. Study Design

The present study was designed to quantify the expression of the active form of these proteins as a measure of DDR activity in operated BC patients, assessing their impact on disease course and clinical outcomes. We focused on a cohort of 222 patients (Table 1), representing a subset of the 750 participants originally enrolled in a multicenter study for the Phase III, prospective, randomized trial of first-generation taxane (GOIM 9902—Gruppo Oncologico Italia Meridionale), led by our study group [21] (Figure 1).

The original trial targeted patients who underwent radical surgery for locally advanced BC and had pathological evidence of axillary lymph node involvement. This study primarily aimed to evaluate the efficacy of integrating a taxane into an anthracycline-based adjuvant chemotherapy regimen. With a median follow-up of 64 months, the trial did not show a statistically significant difference in DFS between the two arms. We attributed such results in part to some study limitations, such as a relatively small sample size and fewer events than expected. In fact, over 94% of participants only had 1-3 positive lymph nodes, suggesting a lower and delayed relapse risk, impacting the lower event rate. Additionally, the epirubicin dose of the EC regimens employed in the other “first-generation” adjuvant trials was lower than the dose employed in our trial, and there is clear evidence of the role of the anthracyclines dose in the adjuvant setting [22].

In the present study, we provide an updated analysis of the endpoints from the original investigation, focusing on the specified 222-patient subset. Additionally, we examine the potential impact of DDR protein expression in tumors for disease relapse and treatment outcomes.

### 2.2. Study Participants

Written informed consent was obtained by all the patients prior to participation in the original study, allowing for the additional analyses performed in the current investigation. This study was conducted in accordance with the Declaration of Helsinki and approved by the Ethics Committees of the IRCCS Regina Elena National Cancer Institute of Rome and IRCCS Istituto Tumori “Giovanni Paolo II”, Bari, Italy. The current study was conducted on women, as implied by the inclusion criteria of the original Phase III study. Patients were considered eligible if they had complete data on clinical features and treatment outcomes. Additionally, sufficient biological material from biopsies or surgical samples was required to test the full panel of antibodies. Tissue microarrays (TMAs) were constructed using samples from 222 histologically confirmed locally advanced breast cancer (BC) patients. These samples were immunohistochemically characterized to evaluate the expression of four DDR kinases (pATR, pATM, pCHK1, and pWEE1) and two DNA damage markers (pRPA32 and γ-H2AX). We assessed the relationship between these biomarkers and their prognostic significance in 208 patients. This assessment was based on the availability of survival outcome information and complete data for the investigated markers. Disease-free survival (DFS) was calculated as the time between the first cycle of chemotherapy until radiological evidence of disease relapse or death due to any cause. Overall survival (OS) was computed as the time from the first cycle of chemotherapy to death due to any cause. 

### 2.3. Immunohistochemistry

The immunohistochemistry (IHC) assessment was carried out on formalin-fixed paraffin-embedded (FFPE) tissue samples. Immunoreactions were revealed by Bond Polymer Refine Detection in an automated autostainer (BondTM Max, Leica Biosystems, Milan, Italy) using the following antibodies: anti-phospho-ATM (Ser1981, clone 7C10D8) mouse monoclonal antibody (Mab) (Rockland, Limerick, PA, USA) at pH 6, antiphospho-ATR (Ser428, clone EPR2184) rabbit Mab (Abcam, Cambridge, UK) at pH 6, anti-phospho-Chk1 (Ser345, clone 133D3) rabbit Mab (Cell Signaling, Danvers, MA, USA) at pH 6, anti-phospho-H2AX (Ser139, clone JBW301) mouse Mab (Millipore, Burlington, MA, USA) at pH 8, anti-phospho-RPA32 (Ser4/Ser8) rabbit polyclonal antibody (Bethyl, Montgomery, TX, USA) at pH 6, and anti-phospho-Wee1 (Ser642, clone D47G5) rabbit Mab (Cell Signaling) at pH 8. Diaminobenzidine was used as the chromogenic substrate. Slides were analyzed by light microscopy, and immunoreactivity was evaluated by two (ADB and CE) investigators independently and blinded to the treatment outcomes. For each antibody, both the percentage of positive neoplastic cells and the score of immunoreaction intensity (0/1+/2+/3+) were evaluated, specifying cellular localization (membrane/nucleus/cytoplasm). In the original study, estrogen receptor (ER) and progesterone receptor (PR) levels were immunohistochemically assessed using an automated autostainer, with a positivity threshold of at least 10% for distinct nuclear immunoreactivity in neoplastic cells. However, for the current investigation, the positivity criterion for ER and PR was revised to include cases where at least 1% of the neoplastic cells demonstrated nuclear immunoreactivity. HER2 protein overexpression was retrospectively analyzed using the DAKO Hercept Test kit based on the intensity and pattern of membrane staining. Additionally, HER2 gene amplification was assessed via FISH in cases with an IHC score of 2+.

### 2.4. K-Means and Hierarchical Clustering Analysis

To comprehensively assess the expression patterns of DDR biomarkers, we performed k-means clustering coupled with hierarchical clustering. K-means clustering was employed to partition the data into distinct groups based on the nuclear and cytoplasmic IRS expression profiles of the DDR biomarkers. This method minimizes the variance within each cluster and ensures that the data points in a cluster are more similar to each other than to those in other clusters.

Hierarchical clustering was subsequently applied to the k-means cluster centroids to further refine the clustering and enhance visualization. This approach provides a dendrogram that illustrates the arrangement of the clusters formed by k-means, allowing for an intuitive understanding of the relationships between different clusters. The resultant heatmap delineates the expression patterns of DDR biomarkers across the identified clusters, facilitating the identification of key biomarker expression profiles and their associations with clinical and pathological data.

### 2.5. Statistical Analysis

#### 2.5.1. Baseline Characteristics and Treatment Allocation

We utilized descriptive statistics to summarize the primary characteristics of the patients and their treatment allocations. Fisher’s exact test and Chi-square tests were employed to compare categorical variables.

#### 2.5.2. Survival Analysis

To estimate and visualize survival probabilities for DFS and OS, we used the Kaplan–Meier (KM) estimator with the results presented in KM curves. Group comparisons were conducted using univariate Cox regression to estimate hazard ratios.

#### 2.5.3. Correlation Analysis between DDR Biomarkers

We calculated Pearson’s correlation coefficients to explore the relationships between the nuclear and cytoplasmic expressions of DDR biomarkers. 

#### 2.5.4. Cox Elastic Net Analysis

We employed an elastic net regularized Cox model to capture nonlinearities and interactions between biomarkers, producing a risk score (RS) for stratifying the study cohort into high-risk and low-risk groups. The elastic net terms and their coefficients were visualized using a circular heatmap representation.

#### 2.5.5. Multivariate Analysis

Univariate and multivariate analyses using the Cox regression model were performed to adjust for potential confounders, including the treatment arm, DDR risk group, tumor stage, HER2 expression levels, and adjuvant treatments. 

The statistical significance was set at a *p*-value of less than 0.05. All analyses and visualizations were conducted using R (version 4.2.3), ensuring robust and reliable data processing and representation.

#### 2.5.6. Validation in TCGA BC Cohort

For validation purposes, we analyzed the TCGA Breast Invasive Carcinoma dataset. We used the median expression value of ATM to stratify patients and compared DFS and OS between the high and low ATM expression groups using the log-rank test and Cox regression analysis.

## 3. Results

### 3.1. Original Study and Baseline Characteristics of Patients

This present study was conducted on a cohort of 222 patients, as a subset of the 750 patients that were included in the original multicenter Phase III prospective randomized trial that was designed and conducted by our study group [21]. In the original trial, eligible patients were 18–70 years old, were treated with radical surgery for locally advanced breast cancer, had histologically proven axillary lymph node involvement, and were unselected for HER2 expression. From April 1999 to October 2005, 374 patients were allocated to Arm A [four cycles of EC (epirubicin 120 mg/m^2^ and cyclophosphamide 600 mg/m^2^ i.v. on day 1 and every 21 days)]; and 376 patients were allocated to Arm B [four cycles of D (docetaxel 100 mg/m^2^ i.v. for 1 h infusion on day 1 and every 21 days, followed by four cycles of EC as above]. No patient was treated with trastuzumab since the evaluation of HER2 at that time was not routinely assessed in the adjuvant setting. At the end of chemotherapy, patients with positive hormonal receptors (ER, PR, or both) were given appropriate hormonal treatment. Radiation therapy was given in cases of conservative surgery or in the case of four or more positive nodes. In the first published analysis, at a median follow-up of 64 months, there was no evidence of disease-free survival (DFS) or overall survival (OS) differences between the control (EC) and the experimental (D → EC) arms, with respective hazard ratios (HRs) of 0.99 (95% CI 0.75–1.31; *p* = 0.95) and 0.84 (CI 0.54–1.31; *p* = 0.45) [21].

In the present analysis, we focused on a subset of 222 patients derived from the original cohort, who were treated at two principal institutions that contributed to the original trial. Specifically, the IRCCS Regina Elena National Cancer Institute (IRE) of Rome, Italy, contributed 88 patients to the current investigation, representing 40% of this subset, while IRCCS Istituto Tumori “Giovanni Paolo II”, Bari, Italy (BARI) accounted for the remaining 134 patients (60%). Key patient characteristics and treatment allocations are reported in Table 1. Regarding HER2 expression, the evaluation was performed subsequently for the present study. 

Fifty-four percent of the patients (119 patients) were assigned to the EC arm and 44% (99 patients) to the D → EC arm, while treatment allocation remained unclear for 2% (4 patients) based on the original study documentation (Figure 1). Adjuvant radiation therapy was administered to 61% of the patients (133 patients), and 74% (160 patients) received adjuvant endocrine therapy.

### 3.2. Updated DFS and OS Analyses

Among 222 analyzed patients, 217 had information regarding treatment allocation and updated survival data. The median follow-up duration was 234 months (interquartile range [IQR], 209.2–257.8 months). In terms of DFS, there was no significant difference between the EC and D → EC treatment arms (hazard ratio [HR] = 1.03, 95% confidence interval [CI]: 0.66–1.6; *p* = 0.913). The 260-month DFS rate was 55% in the EC arm compared to 57% in the D → EC arm (Figure 2A). Similarly, OS did not significantly differ between patients receiving EC and those receiving D → EC (HR = 1.15, 95% CI: 0.72–1.84, *p* = 0.547). The 260-month OS rates were 67% for the EC arm and 60% for the D → EC arm (Figure 2B). Moreover, we assessed the treatment effect in the subgroup of patients (N = 55) with HER2-positive disease separately (Figure 2C,D). 

There was a trend for longer DFS (HR = 0.51, 95% CI: 0.23–1.1, *p* = 0.084) and OS (HR = 0.7, 95% CI: 0.32–1.54, *p* = 0.369) in patients treated with D → EC. With regard to DFS in this subgroup of patients, the 260-month rate was 29% and 58%, respectively, for the EC and D → EC arms. The 260-month OS rate was 50% in the EC arm and 58% in the D → EC arm. The same analysis was performed on the remaining part of the cohort (HER2-negative disease) and did not show any trend among the two treatment arms (Appendix A). In the overall analyzed cohort, the percentage of patients that received D → EC was 49.2% and 42.9%, respectively, in patients with HER2-positive disease and HER2-negative disease, with no statistically significant difference (Fisher’s exact test *p* = 0.43; Appendix A).

In our analysis, the two most impactful prognostic factors for both DFS and OS were HER2 expression and cancer stage (Appendix A). We observed a significant variation in DFS and OS across different levels of HER2 expression: overall, patients with HER2-null disease (IHC score 0) had the best outcomes, followed by those with HER2-low disease (IHC score of 1+ or 2+ ISH negative), and the worst outcomes were seen in patients with HER2-positive disease (IHC score of 3+ or 2+ ISH positive) (Appendix A). This result was statistically significant, as indicated by the log-rank test *p*-values of 0.024 for DFS and 0.028 for OS. Likewise, cancer staging also showed a clear correlation with survival outcomes. Patients with stage 2 disease exhibited better DFS and OS compared to those diagnosed with stage 3 disease (log-rank test *p*-values of 0.0032 for DFS and 0.00025 for OS) (Appendix A).

### 3.3. DDR Biomarker Expressions and Correlation with Clinical–Pathological Characteristics

The raw nuclear (_N) and cytoplasmic (_C) expression levels of ATM, ATR, CHK1, H2AX, RPA32, and WEE1 phosphoproteins were converted into a standardized immune-reactive score (*IRS*) ranging from 0 to 12 for each marker. This scoring system allowed for further categorization into four distinct expression profiles: “positive, strong” (IRS 9–12), “positive, intermediate” (IRS 4–8), “positive, weak” (IRS 2–3), and “negative” (IRS 0–1) [23,24] (Figure 3A–D). As expected, most biomarkers exhibited higher expression in the nucleus compared to the cytoplasm, except for CHK1, which was predominantly cytoplasmic. Specifically, ATM, ATR, and RPA32 showed the highest nuclear expression, with respective levels of 58%, 77%, and 58% classified as “positive, strong” and “positive, intermediate” (Appendix A). We further explored the pairwise linear associations between all measured biomarker levels both in the nucleus and in the cytoplasm by calculating the respective Pearson’s correlation coefficients (Figure 3E). A marked positive correlation was observed in nuclear expressions, particularly for ATR, ATM, and RPA32, with *r* coefficients ranging from 0.47 to 0.53, supported by highly significant *p*-values (<0.001).

To comprehensively assess the expression patterns, we performed k-means coupled with hierarchical clustering analyses using the 12 nuclear/cytoplasmic IRS expression profiles of the DDR biomarkers. The resultant heatmap (Figure 4) delineated three main DDR biomarker k-means clusters and two patient k-means clusters.

Concerning biomarker expression profiles, the first k-means cluster included high-expression nuclear phosphoproteins (ATR_N, RPA32_N, and ATM_N), while the second cluster consisted of biomarkers with intermediate expression levels, both nuclear and cytoplasmic (CHK1_C, WEE1_N, and H2AX_N). The third cluster encompassed mainly cytoplasmic biomarkers with lower expression. Regarding patient k-means clusters, the first cluster (upper part of the heatmap) included patients with high DDR biomarker expression, while the second cluster (lower part) comprised patients with low expression. Within each cluster, further hierarchical clustering was performed based on DDR profiles. The central region of the heatmap predominantly represented cases with a lower DDR from both k-means clusters. In addition to the DDR biomarker expression heatmap, clinical, pathological, and molecular data of each patient and their tumors are annotated in Figure 4. In the high DDR k-means cluster, cases with the highest DDR expression according to hierarchical ordering tended to have tumors with greater anatomical extension, indicated by higher stages and increased lymph node involvement. However, this finding is difficult to interpret since it only represents the k-means high category and not the entire population. Moreover, this subgroup of tumors showed a tendency for lower HR expression and higher HER2 expression. Regarding the relationship between DDR markers and proliferation markers, such as Ki67, the high DDR expression cluster did not show significant differences in Ki67 levels compared to the lower DDR expression cluster. This suggests that DDR marker expression does not directly correlate with proliferation rates, as indicated by Ki67. Additionally, there were no clear distribution differences in terms of disease-free survival (DFS) and overall survival (OS) times and their respective censoring rates. Overall, this figure serves primarily as a visualization tool to help explore potential relationships between DDR markers and various tumor and patient-related features. This type of visualization is not intended for statistical testing and should be interpreted with caution. 

### 3.4. Identification of Prognostic DDR Groups through Cox Elastic Net

To discern the influence of DDR biomarkers on DFS and OS, we chose to consider the collective effect of their expression profile through a single linear predictor as opposed to testing biomarkers singularly. To capture nonlinearities and interactions between biomarkers, we employed first-degree and second-degree terms in the predictor and used an elastic net-regularized Cox model to reduce overfitting and capture the most impactful coefficients (see Section 2). Using this technique, we produced a risk score (RS), which allowed us to stratify the study cohort into patients with high-risk disease and patients with low-risk disease. The most relevant contributors to this RS are displayed in Figure 5A. Nuclear ATM and nuclear ATR were identified as the principal negative effectors influencing the RS, distinguishing themselves as key biomarkers. Furthermore, nuclear ATM and nuclear ATR negatively impacted the RS through mutual interactions (ATM_N*ATR_N) and by interacting with the remaining biomarkers (Figure 5A, the complete list of the elastic net terms is displayed in Appendix A). We subsequently analyzed the expression levels of each biomarker across the two risk groups, observing that nuclear ATM and nuclear ATR exhibited the most significant differential expression with higher levels in the low-risk category (Figure 5B). 

The survival analysis of the entire cohort by stratifying according to these two risk groups is shown in Figure 6A,B. Patients classified in the low-risk disease groups demonstrated a 59% decrease in relapse risk (HR = 0.41, 95%CI: 0.25–0.65, *p* < 0.001) and a 39% reduction in death risk (HR = 0.61, 95% CI: 0.38–0.99, *p* = 0.0.046) (Figure 6A,B). In addition, to evaluate the predictive potential of the DDR risk groups, we analyzed the efficacy of treatment arms separately within high-risk and low-risk categories (Figure 6C,D and Appendix A). 

It is noteworthy that no significant difference was observed in terms of DFS or OS within the high-risk group across different treatment arms, as shown in Figure 6C,D, respectively. This lack of difference in outcomes was similarly observed in the low-risk group, with DFS and OS (Appendix A), excluding the predictive value of the risk groups as previously defined.

As an example, which helps to visualize the IHC quantification of biomarkers, we presented histological slides from two patients, highlighting the nuclear expressions of ATR and ATM as the two main components of the risk classifier. Patient 212, a pre-menopausal woman with a triple-negative subtype diagnosed at stage 2 and treated with EC, exhibited strong positive immune reactivity for both phosphorylated ATR and ATM in the tumor (Figure 7A,B), correlating with a DFS of 184 months. Conversely, patient 162, also pre-menopausal but with a luminal B HER2-negative subtype, diagnosed at stage 3 and treated with D → EC, showed negative nuclear expressions for both ATR and ATM (Figure 7C,D), correlating with a DFS of 74 months.

### 3.5. Multivariate Models for Treatment Arm and DDR Risk Group Effects on DFS and OS

To dissect the individual effect of the treatment arm and DDR risk group stratification on DFS and OS while concurrently adjusting for other potential confounding factors, we built multivariate models for both DFS and OS. Besides variables representing the treatment arm and DDR risk group, we also included other determinants identified as influential on DFS or OS, specifically tumor stage and HER2 expression levels (as detailed in Section 3.2) and adjuvant hormonal treatment. The latter, also serving as a proxy for tumor hormone receptor expression positivity, demonstrated a significant reduction in the risk for both DFS (HR = 0.53, 95% CI: 0.33–0.85, *p* = 0.009) and OS (HR = 0.52, 95% CI: 0.32–0.84, *p* = 0.008). Given its clinical significance, we also incorporated a variable representing adjuvant radiation therapy in the multivariate models for DFS and OS. Despite its lack of statistical significance in univariate analyses for DFS (HR = 0.75, 95% CI: 0.47–1.20, *p* = 0.22) and OS (HR = 0.68, 95% CI: 0.42–1.10, *p* = 0.11), its inclusion was warranted due to its expected inverse correlation with disease stage as determined by the inclusion criteria of patients in the study (all stage three patients had to undergo radiation therapy).

In the resultant multivariate analysis for DFS (Figure 8A), the treatment arm did not exhibit a statistically significant impact (HR for EC vs. D → EC (as reference) = 0.97, 95% CI: 0.60–1.58, *p* = 0.908). The only variables demonstrating an independent effect were tumor stage (HR for stage 3 vs. stage 2 (as reference) = 2.47, 95% CI: 1.47–4.14, *p* = <0.001) and DDR risk category (HR for low-risk vs. high-risk groups (as a reference) = 0.19, 95% CI: 0.19–0.53, *p* = <0.001). Concerning OS (Figure 8B), the multivariate analysis corroborated the negligible effect of the treatment arm (HR for EC vs. D → EC (as a reference) = 0.93, 95% CI: 0.56–1.56, *p* = 0.791), while independently significant variables included tumor stage (HR for stage 3 vs. stage 2 (as a reference) = 3.54, 95% CI: 1.98–6.33, *p* < 0.001), adjuvant hormonal treatment (HR for recipients vs. non-recipients (as a reference) = 0.52, 95% CI: 0.29–0.94, *p* = 0.031), and adjuvant radiation therapy (HR for recipients vs. non-recipients (as a reference) = 0.60, 95% CI: 0.35–1.01, *p* = 0.056).

### 3.6. Evaluating the Validation of the DDR Prognostic Effect in the TCGA BC Cohort

In our endeavor to externally validate findings relative to the influence of DDR biomarkers on DFS and OS, we utilized clinical and proteomic data from the Breast Invasive Carcinoma—TCGA, Pancancer Atlas dataset [25]. The dataset encompassed 1084 breast cancer patients, with primary tumor protein expression profiling data accessible for 876 cases. Given the limited availability of phosphoprotein site level expression data (present in only 10% of the cohort), we opted for Reverse Phase Protein Array (RPPA) normalized expression levels for our analysis. We focused on a subset of patients who underwent upfront radical surgery. Metadata did not provide information on the adjuvant treatments. Our goal was to validate the prognostic effect of the DDR biomarkers, as identified in our internal cohort. However, among the RPPA expression profiles, only two proteins, ATM and CHK1, corresponded to our selected DDR biomarkers (ATM, ATR, CHK1, RPA32, H2AX, and WEE1). Realizing the limitation in fully validating our DDR risk score, we nonetheless proceeded to examine the impact of ATM expression, a key component of our prognostic score, demonstrating a protective effect for both DFS and OS. We stratified the selected TCGA cohort based on the median ATM expression value, dividing patients into high and low ATM expression groups. Disease-free survival data were available for 750 patients, and OS data were available for 848 patients. The subsequent comparison of DFS and OS between these groups revealed that patients with low ATM-expressing tumors exhibited a significantly higher risk of relapse (HR for DFS = 1.67, 95% CI: 1.01–2.78, *p* = 0.044) and higher risk of death (HR for OS = 1.43, 95% CI: 0.97–2.1, *p* = 0.07) (Appendix A).

## 4. Discussion

Our investigation, examining a cohort derived from a multicenter Phase III randomized trial encompassing patients with operated early/locally advanced BC not selected based on molecular subtype, provides relevant insights into the intricate factors influencing clinical outcomes in this disease setting. This study is particularly significant in the historical context preceding the widespread adoption of anti-HER2 therapies in early treatment settings. We believe that this backdrop allows for a nuanced understanding of how treatment strategies, DDR biomarkers, and clinical–pathological characteristics impact patient outcomes.

In this extended 234-month follow-up, we observed no statistically significant differences in DFS and OS between the EC and D → EC treatment arms. This result aligns with the initial findings of the trial [21] yet stands in contrast to the extensive body of research that has established the anthracycline–taxane sequence as the standard of care in this context [26]. An additional 14 years of follow-up, compared to the previously published results, permitted an increase in the DFS and OS event rates, albeit within the constraints of a limited subset compared to the original study population. Indeed, the subset of patients that we analyzed did not have the original study’s statistical power to detect nuanced differences in DFS and OS. Furthermore, the complexity of the findings was heightened by the diversity in disease subtypes within our patient cohort, compounded by the absence of anti-HER2 therapy in the management of the HER2-positive BC subtype. Additionally, the high epirubicin dose in the control arm and the employment of docetaxel as the taxane in the D → EC group could have partly influenced the outcomes. This aspect has its own relevance under the context of the substantial body of research that identifies paclitaxel in a dose-dense schedule as the preferable option in such settings, primarily due to its more favorable toxicity profile, which typically ensures greater treatment adherence and possibly also some advantage in clinical outcomes [27,28]. Despite these multifaceted considerations, our study does underscore a trend regarding the therapeutic benefits conferred by the addition of docetaxel in the HER2-positive BC subtype without having received trastuzumab. In this numerically limited disease subset, we noted a numerical but not statistically significant enhancement of DFS and OS with the inclusion of docetaxel in adjuvant treatment over the EC treatment. This observation may echo prior studies hinting at a predictive association between taxane responsiveness and HER2-positivity. However, it is important to note that this aspect continues to be a subject of considerable controversy in the field [29].

Our study’s observation, which revealed the significant impact of HER2 expression on DFS and OS at three levels, holds a particular interest in the current context of HER2-positive BC developments. Specifically, we observed progressively improved outcomes across HER2-null, HER2-low, and HER2-positive disease categories in terms of both DFS and OS. These findings are particularly insightful, considering that the patient cohort was treated before the adoption of anti-HER2 therapies in the adjuvant setting. This scenario provides clear confirmation of the inherent prognostic impact of HER2 status [30] and emphasizes the importance of recognizing HER2-low as a separate prognostic group. Additionally, the relevance of these results is heightened when compared with more recent data from the era of HER2-targeted agents. Notably, an HER2-low status has also emerged as a predictive factor for the newer generation anti-HER2 agents [31]. Contemporary studies, where HER2-positive and, more recently, HER2-low disease were treated with anti-HER2 agents, show a reverse trend in DFS and OS. In this new context, HER2-positive status correlates with the most favorable survival, followed by HER2-low and HER2-null diseases [32]. The juxtaposition of these findings marks the profound prognostic and predictive implications of HER2 expression in BC, particularly considering the divergent trajectories observed in the absence and presence of anti-HER2 therapies. This divergence further highlights the essential role of such agents in the management of HER2-positive BC, dependent on an accurate assessment of HER2 expression levels.

Among complementary adjuvant treatments, we confirmed the advantage conferred by both endocrine treatment and radiation therapy, particularly in terms of OS, although the former is confounded by the underlying positive prognostic effect of hormonal receptor expression.

Our study also provided some relevant novel insights on the role of DDR expression in BC. We used the phosphoprotein profile to evaluate DDR activation, recognizing it as a reliable indicator of BC’s biological characterization, disease progression, and treatment responsiveness. Notably, DDR’s complex pathological processes often transcend single-gene alterations and may not be fully reflected at the genomic level. Initial analyses indicated a strong positive correlation between DDR biomarkers, especially ATM and ATR nuclear expressions, highlighting their intertwined role in the DDR pathway. This was reinforced by the results from risk stratification based on elastic net Cox modeling, which identified ATM and ATR as key influencers in DDR overall profile variation. Interestingly, a DDR risk predictor predominantly and inversely driven by ATM and ATR expressions emerged, offering a risk score that independently classified our cohort based on prognosis. Contrary to some previous studies–including those from our research group—indicating a negative prognostic impact of increased ATM-ATR expression [33,34,35,36], the current findings reveal an inverse relationship with this risk score, suggesting a protective effect in terms of better DFS and OS in BC. The results of the present study, however, align with additional research demonstrating similar positive prognostic implications of elevated ATM expression levels [12,37,38]. Furthermore, our validation in the TCGA cohort confirmed ATM’s protective role. Nevertheless, interpreting the prognostic and predictive value of ATM, ATR, and other DDR components remains challenging due to conflicting results across heterogeneous disease backgrounds and treatment settings, including neoadjuvant, metastatic, and adjuvant therapies [39]. The role of DDR in cancer biology appears context-dependent, either indicating tumor vulnerability or resilience. Subtype-specific DDR roles in BC add another layer of complexity [40]. Additionally, distinct DDR expressions in stromal versus cancer cells necessitate the precise evaluation of accurate prognostic interpretations [41,42]. Moreover, the genomic backdrop is crucial in understanding DDR deregulation, as concurrent genomic alterations might significantly alter DDR-related outcomes [43,44]. The emerging development of DDR inhibitors further highlights the importance of accurately framing ATM, ATR, and other DDR components within BC’s biological and clinical context [45,46].

Expanding the landscape of biomarkers predictive of prognosis in breast cancer, other pathways, such as histaminergic and dopaminergic systems, have also garnered attention. For instance, a study investigating the role of histamine and histamine H4 receptor (H4R) ligands in a TNBC model found that increased histidine decarboxylase (HDC) gene expression is associated with better relapse-free and overall survival in breast cancer patients [47]. Histamine treatment in a TNBC model reduced tumor growth and increased apoptosis, highlighting the complex role of histamine and its potential as a therapeutic agent in breast cancer. Furthermore, the HDC expression level is suggested to have prognostic value in breast cancer. Similarly, the dopaminergic system, highlighted in recent studies, indicates that dopamine receptors, particularly DRD2 and DRD3, are upregulated in breast cancer, potentially contributing to tumor growth and progression. Conversely, DRD5 expression is often downregulated, which might influence tumor suppression mechanisms [48]. Our findings on DDR biomarkers, specifically ATM and ATR, suggest a protective role, which is concordant with the findings relative to the histaminergic system but contrasts with some of the detrimental effects seen with dysregulated dopaminergic signaling. These differences underscore the multifaceted nature of breast cancer biology, where various signaling pathways intersect and influence disease outcomes. Future research should aim to explore these intersections, potentially unveiling novel therapeutic targets that modulate multiple pathways simultaneously. As previously discussed in this section, our study operates within defined constraints. Conversely, the study’s strengths lie in its longitudinal scope, with a 234-month follow-up period, offering a rare, long-term perspective on disease relapse and treatment efficacy. Our cohort, although a subset of the original study, benefits from the robustness imparted by the multicenter, Phase III randomized trial design, which enhances the generalizability of our results. Particularly notable is the insight provided into the prognostic impact of HER2 expression across different subtypes in a pre-anti-HER2 therapy context. Our findings also contribute novel insights into DDR expression in early BC, particularly the roles of ATM and ATR, offering a new prognostic lens. The validation of these findings in an independent TCGA cohort, although partial, further solidifies our conclusions, underscoring the nuanced yet significant role of DDR in BC biology and its possible treatment responsiveness.

## 5. Conclusions

This study provides valuable insights into the prognostic significance of DDR components and HER2 expression in early BC, particularly in a pre-anti-HER2 therapy setting. Despite no significant differences in DFS and OS between the treatment arms, a nuanced understanding of the protective implications of DDR markers such as ATM and ATR and taxane responsiveness linked to HER2 status highlight their potential roles in refining BC treatment strategies. In conclusion, our findings advocate for a comprehensive, personalized patient care approach in BC, which is particularly relevant amidst the dynamic landscape of cancer therapies. The integration of molecular markers like DDR biomarkers with clinical and treatment factors promises more effective, tailored treatment strategies, which can be adaptable to both historical and emerging therapeutic contexts.

## Figures and Tables

**Figure 1 cancers-16-02628-f001:**
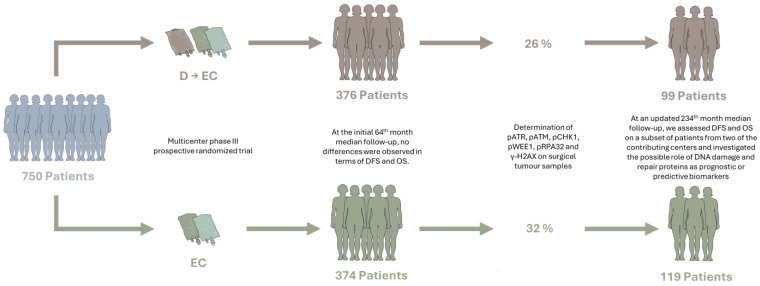
Flowchart of patient enrollment and follow-up in the original Phase III trial and current study. EC = epirubicin-cyclophosphamide; D = docetaxel; DFS = disease-free survival; and OS = overall survival.

**Figure 2 cancers-16-02628-f002:**
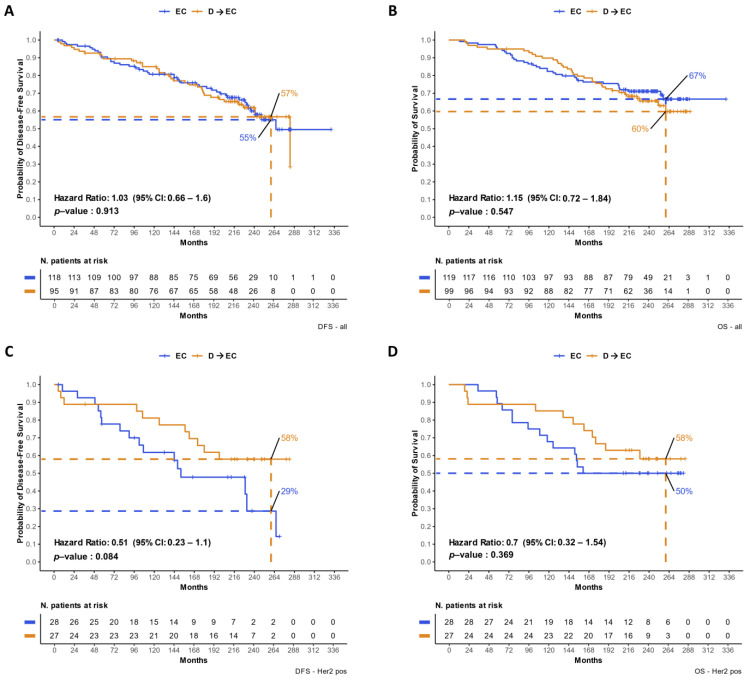
Updated survival analyses at 234 months of follow-up. Panels (**A**,**B**) illustrate the KM curves for DFS and OS, respectively, comparing the EC treatment arm to the D → EC treatment arm across the entire subset of patients included in the study. Panels (**C**,**D**) demonstrate the DFS and OS comparisons’ specifically, respectively, in patients with HER2-positive disease. KM = Kaplan–Meier; DFS = disease-free survival; OS = overall survival; EC = epirubicin-cyclophosphamide; D = docetaxel; and HR = hazard ratio.

**Figure 3 cancers-16-02628-f003:**
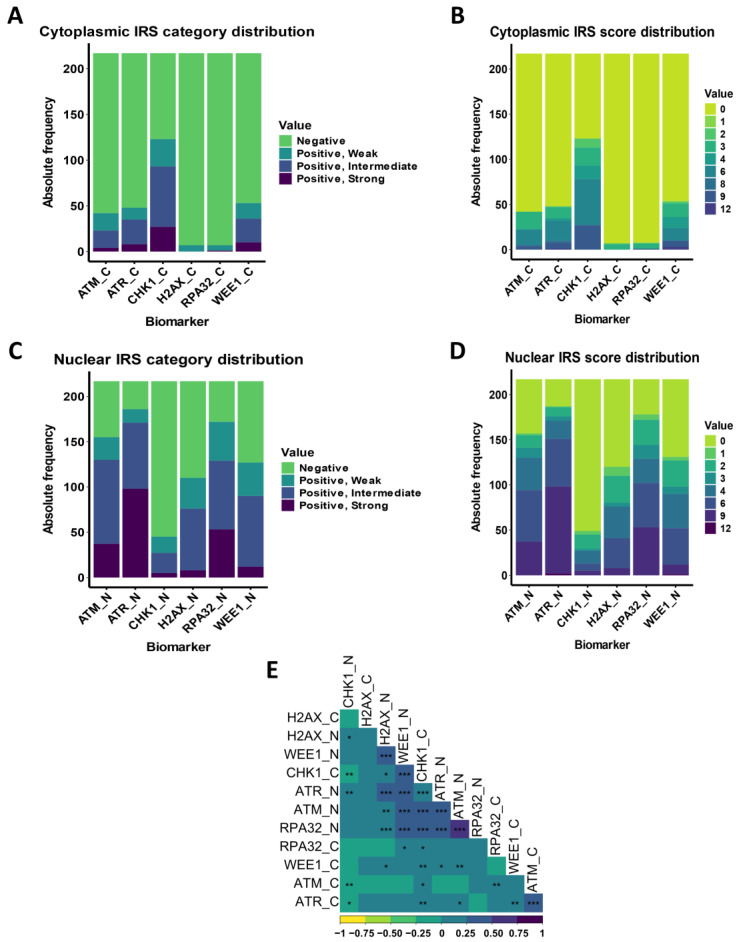
Overall expression and correlations of DDR biomarkers. Panels (**A**–**D**) show the distribution of immunohistochemical expression levels for DDR biomarkers. Panels (**A**,**B**) display the cytoplasmic expression in terms of IRS categories and IRS scores. Panels (**C**,**D**) represent the nuclear expression in terms of IRS categories and IRS scores. Panel (**E**) depicts a correlation matrix for the DDR biomarkers based on their IRS scores of both nuclear (denoted by ‘_N’) and cytoplasmic (denoted by ‘_C’) expressions, with color intensity and star symbols indicating the strength and significance of the correlations: “*” for *p* < 0.05, “**” for *p* < 0.01, and “***” for *p* < 0.001. IRS = immunoreactive score.

**Figure 4 cancers-16-02628-f004:**
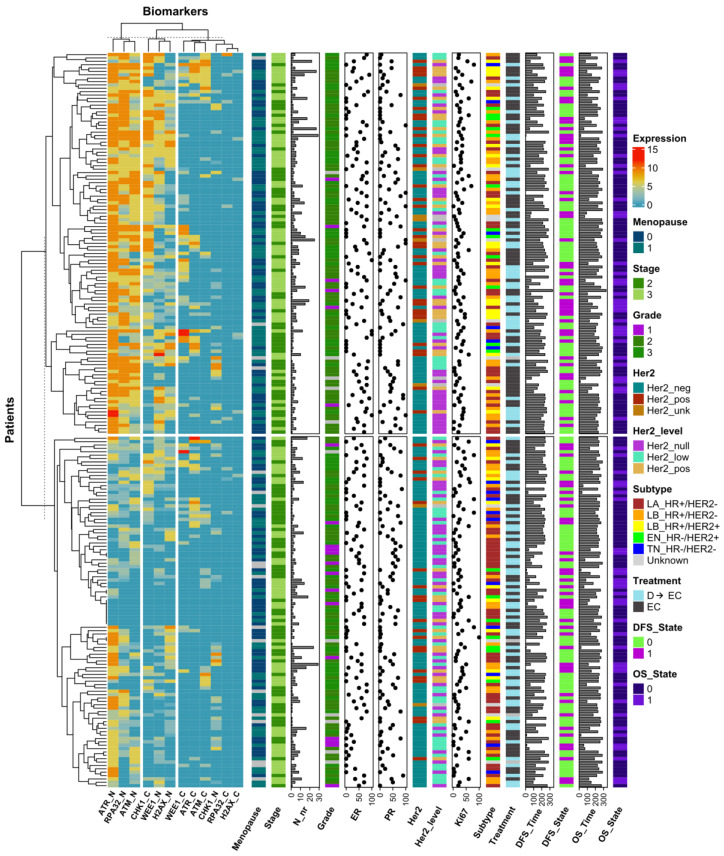
Heatmap of DDR biomarker expression and patient characteristics with clustering analysis. This heatmap visualizes the primary k-means clustering, reflected by the distinct splits in the patient groups, and secondary distance-based hierarchical clustering, which orders patients and DDR biomarkers within each subgroup, as indicated by the dendrogram to the left and to the top. It displays the expression levels of DNA damage response (DDR) biomarkers within the nuclear and cytoplasmic compartments across the cohort of patients. The right side of the heatmap is annotated with various patient characteristics: ‘Menopause’ status, ‘Stage’ of cancer, ‘Grade’ of the tumor, classical ‘HER2’ positivity/negativity, ‘HER2_level’ indicating HER2-positive, HER2-lowe, and HER2-null statuses, the ‘ER’ percentage representing estrogen receptor expression, the ‘PR’ percentage for progesterone receptor expression, the proliferation marker ‘Ki67’, breast cancer ‘Subtype’, ‘Treatment’ received (D → EC or EC), ‘Disease-Free Survival’ (DFS), event occurrence in DFS (‘DFS_State’), ‘Overall Survival’ (OS), and event occurrence in OS (‘OS_State’).

**Figure 5 cancers-16-02628-f005:**
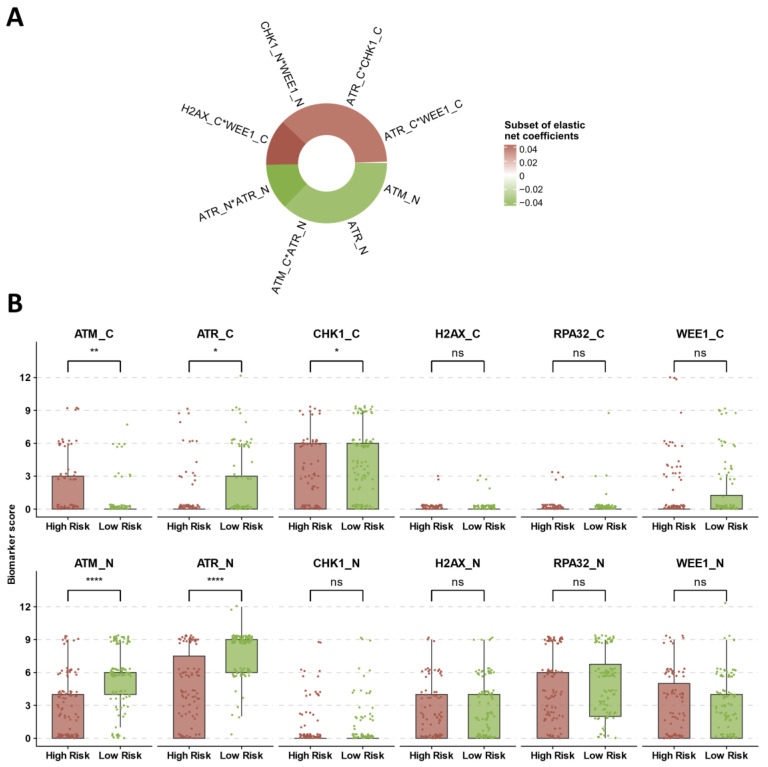
Cox elastic net risk predictor and biomarker distribution based on risk groups. Panel (**A**) displays the most relevant direct and interaction terms relative to the effect of the biomarkers in the Cox elastic net predictor. Panel (**B**) shows the distribution of the biomarkers according to the risk groups. Significance levels: “ns” for *p* ≥ 0.05, “*” for *p* < 0.05, “**” for *p* < 0.01, and “****” for *p* < 0.0001.

**Figure 6 cancers-16-02628-f006:**
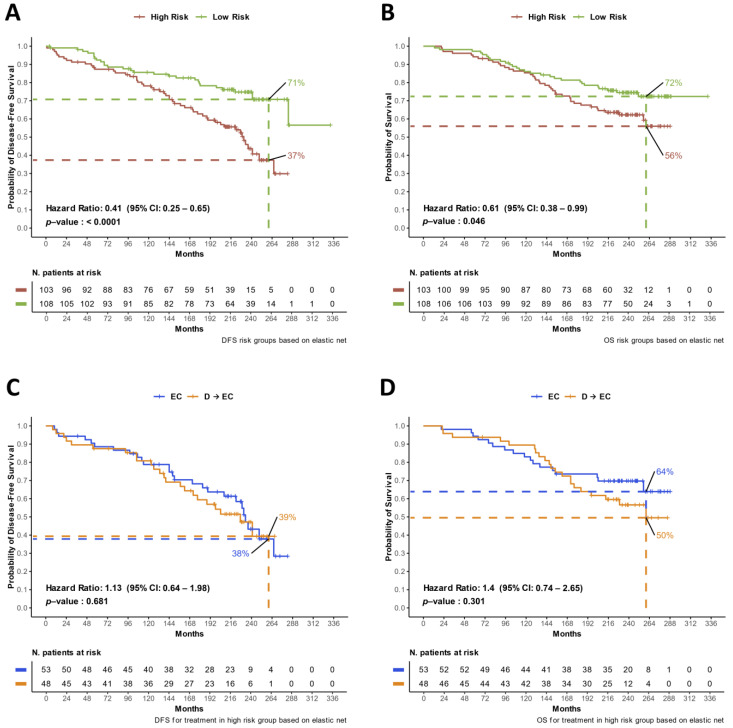
Prognostic and predictive potential of the DDR risk score. Panels (**A**,**B**) illustrate the KM curves for DFS and OS, respectively, comparing the high-risk versus the low-risk groups across the entire subset of patients included in the study. Panels (**C**,**D**) demonstrate the DFS and OS comparisons, respectively, between the EC treatment arm and the D → EC treatment arm, specifically in patients with high-risk disease. HR = hazard ratio.

**Figure 7 cancers-16-02628-f007:**
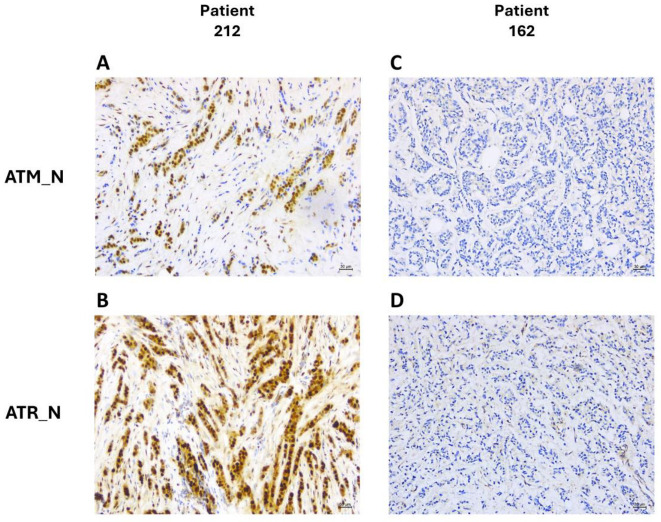
Showcasing the immunohistochemical assessment of ATM and ATR expressions. The figure presents the comparative immunohistochemical staining of DDR biomarkers ATM and ATR in the nucleus (denoted as ATM_N and ATR_N) from two distinct patient cases. Patient_212 (upper images), who received EC treatment, was premenopausal, and diagnosed with stage 2 triple-negative breast cancer. Panels (**A**,**B**) exhibit the strong, positive nuclear expressions of ATM and ATR, respectively. This patient has a documented disease-free survival (DFS) and overall survival (OS) of 184 months. Patient_162 (lower images), who received D → EC treatment, was premenopausal with stage 3 Luminal B HER2-negative breast cancer. Panels (**C**,**D**) show negative nuclear staining for both ATM and ATR. These contrasting profiles highlight the variance in DDR biomarker expressions across different patient treatments and cancer subtypes.

**Figure 8 cancers-16-02628-f008:**
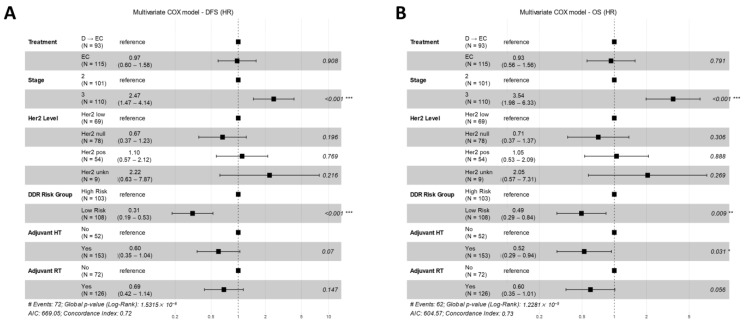
Multivariate analysis of predictor factors for DFS and OS. Panel (**A**) presents a forest plot illustrating the results from the multivariate analysis for disease-free survival (DFS) across the study cohort. The plot includes hazard ratios for various factors such as treatment type (EC or D → EC), cancer stage, HER2 level, DDR risk group, and the receipt of adjuvant hormone therapy (HT) or radiation therapy (RT). Statistical significance is denoted by the distance of the confidence interval from the reference line (hazard ratio of 1), and the *p*-value is on the right. Panel (**B**) shows a similar forest plot for overall survival (OS). Significance levels: “*” for *p* < 0.05, “**” for *p* < 0.01, and “***” for *p* < 0.001.

**Table 1 cancers-16-02628-t001:** Baseline characteristics of patients in the study.

Feature	Nr. (%)
**Age**	Median, 51
	Range, 28–70
**Centre**	
*IRE*	88 (40)
*BARI*	134 (60)
**Menopause**	
*Pre-Menopause*	100 (45)
*Post-Menopause*	104 (47)
*Unknown*	18 (8)
**Stage**	
*2*	106 (48)
*3*	116 (52)
**Subtype**	
*Luminal A HR+/HER2−*	83 (37)
*Luminal B HR+/HER2−*	53 (24)
*Luminal B HR+/HER2+*	33 (15)
*HER2 Enriched (HR−/HER2+)*	22 (10)
*Triple Negative (HR−/HER2−)*	18 (8)
*Unknown*	13 (6)
**Her2 Level**	
*HER2 positive*	55 (25)
*HER2 low*	70 (32)
*HER2 null*	87 (39)
*Uknown*	10 (4)
**Treatment Arm**	
*EC*	119 (54)
*D → EC*	99 (44)
*Uknown*	4 (2)
**Adjuvant RT**	
*No*	75 (34)
*Yes*	133 (61)
*Unknown*	10 (5)
**Adjuvant HT**	
*No*	55 (25)
*Yes*	160 (74)
*Unknown*	3 (1)

Nr. = number, HR = hormone receptors, HER2 = human epidermal growth factor receptor 2, EC = epirubicin-cyclophosphamide; D = docetaxel; RT = radiation therapy; and HT = hormone therapy.

## Data Availability

The internal dataset generated and analyzed during the current study is available in the GARRbox repository [link: https://gbox.garr.it/garrbox/s/J8JrcIFa28ICOiw]. The repository of the external validation dataset and the R code utilized to perform the analysis, and a brief description of this study is available on GitHub [link: https://github.com/bbdataeng/PHOBOS].

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
