# Peer review of "DNA Damage Response in Early Breast Cancer: A Phase III Cohort in the Phobos Study"

_cancers, 2024, doi:10.3390/cancers16152628_

Round 1

Reviewer 1 Report

Comments and Suggestions for Authors

The manuscript is presented very well. The illustrative material presented is informative and very well integrated into the results.

In general, there are no comments.

Author Response

Thank you very much for taking the time to review this manuscript. Please find the detailed responses below

Comment 1: The manuscript is presented very well. The illustrative material presented is informative and very well integrated into the results. In general, there are no comments.

Response to Comment 1: Thank you for your positive feedback and for acknowledging the quality and integration of the illustrative material. We are pleased that you found the manuscript well-presented and informative.

Reviewer 2 Report

Comments and Suggestions for Authors

1. In Figure 1 it is necessary to show at what stage of the study and in what proportion of patients 4 DDR kinases (pATR, pATM, pCHK1, and pWEE1) and 2 DNA damage markers (pRPA32 and γ-H2AX) were determined. Please add this stage to the diagram.

2. Typo in table 1 - HER2 Enriched (HR+/HER2+), needs to be corrected to HR-/HER2+.

3. Lines 205-230 I repeat the information given in table 1. For what purpose is the information duplicated?

4. Figure 2 very small drawings and inscriptions, almost unreadable. I recommend redoing it and dividing it into 2 drawings: survival indicators (1) and DDR markers (2).

5. Figure 4 is also unreadable: make Figures A and B vertical and increase the font size for the inscriptions. Do the same for Figure 6.

6. Figure 3 contains a lot of information, but it cannot be analyzed. In the text, I would like to see an explanation of how the levels of DDR markers are related, for example, to the proliferation marker and other indicators.

Author Response

Thank you very much for taking the time to review this manuscript. Please find the detailed responses below and the corresponding revisions in the re-submitted files.

Comment 1: In Figure 1 it is necessary to show at what stage of the study and in what proportion of patients 4 DDR kinases (pATR, pATM, pCHK1, and pWEE1) and 2 DNA damage markers (pRPA32 and γ-H2AX) were determined. Please add this stage to the diagram.

Response to comment 1: Thank you for your insightful comment. We have revised Figure 1 to include the proportion of patients in which the 4 DDR kinases (pATR, pATM, pCHK1, and pWEE1) and 2 DNA damage markers (pRPA32 and γ-H2AX) were determined. Additionally, we have indicated the stage at which these determinations were performed. We believe these modifications enhance the clarity and informativeness of the figure.

Comment 2. Typo in table 1 - HER2 Enriched (HR+/HER2+), needs to be corrected to HR-/HER2+.

Response to comment 2: Thank you for pointing out the typo in Table 1. We have corrected the HER2 Enriched category to HR-/HER2+ as you suggested. We appreciate your attention to detail.

Comment 3. Lines 205-230 I repeat the information given in table 1. For what purpose is the information duplicated?

Response to comment 3: Thank you for your valuable feedback. We have reviewed the manuscript and agree that the information in Lines 205-230 duplicates what is already presented in Table 1. To avoid redundancy and improve clarity, we have removed the repetitive sentences in the text while retaining the information in Table 1.

Comment 4. Figure 2 very small drawings and inscriptions, almost unreadable. I recommend redoing it and dividing it into 2 drawings: survival indicators (1) and DDR markers (2).

Response to comment 4. Thank you for your suggestion regarding Figure 2. We have followed your recommendation and divided the original figure into two separate figures for better readability. The first part, now Figure 2, displays only the survival curves. The second part, now Figure 3, contains only the DDR expression and correlations. We have also slightly increased the font size to enhance clarity.

Comments 5. Figure 4 is also unreadable: make Figures A and B vertical and increase the font size for the inscriptions. Do the same for Figure 6.

Response to comments 5: Thank you for your feedback on Figures 4 and 6. We have made the modifications as you suggested. Figures A and B in Figure 4, now renumbered as Figure 5 due to the splitting of the original Figure 2, are arranged vertically with increased font size for the inscriptions. Similarly, Figure 6, now renumbered as Figure 7, has been adjusted with the same improvements. These changes have made both figures clearer and more understandable.

Comment 6. Figure 3 contains a lot of information, but it cannot be analyzed. In the text, I would like to see an explanation of how the levels of DDR markers are related, for example, to the proliferation marker and other indicators.

Response to comment 6: Thank you for your comment regarding Figure 3, now renumbered as Figure 4 due to the splitting of the original Figure 2. We have revised the text to provide a clearer explanation of how the levels of DDR markers are related to proliferation markers and other indicators. The analysis performed aimed to help visualization and provide a high-level descriptive insight, not to test specific differences. Please, find in the manuscript the following updated description. "To comprehensively assess the expression patterns, we performed k-means coupled with hierarchical clustering analyses using the 12 nuclear/cytoplasmic IRS expression profiles of the DDR biomarkers. The resultant heatmap (Figure 4) delineated three main DDR biomarker k-means clusters and two patient k-means clusters. Concerning biomarker expression profiles, the first k-means cluster included high-expression nuclear phosphoproteins (ATR_N, RPA32_N, ATM_N), while the second cluster consisted of biomarkers with intermediate expression levels, both nuclear and cytoplasmic (CHK1_C, WEE1_N, H2AX_N). The third cluster encompassed mainly cytoplasmic biomarkers with lower expression. Regarding patient k-means clusters, the first cluster (upper part of the heatmap) included patients with high DDR biomarker expression, while the second cluster (lower part) comprised patients with low expression. Within each cluster, further hierarchical clustering was performed based on DDR profiles. The central region of the heatmap predominantly represented cases with lower DDR from both k-means clusters. In addition to the DDR biomarker expression heatmap, clinical, pathological, and molecular data of each patient and their tumors were annotated in Figure 4. In the high DDR k-means cluster, cases with the highest DDR expression according to hierarchical ordering tended to have tumors with greater anatomical extension, indicated by higher stage and increased lymph node involvement. However, this finding is difficult to interpret since it only represents the k-means high category and not the entire population. Moreover, this subgroup of tumors showed a tendency for lower HR expression and higher HER2 expression. Regarding the relationship between DDR markers and proliferation markers such as Ki67, the high DDR expression cluster did not show significant differences in Ki67 levels compared to the lower DDR expression cluster. This suggests that DDR marker expression does not directly correlate with proliferation rates as indicated by Ki67. Additionally, there were no obvious distribution differences in terms of DFS and OS times, and their respective censoring rates. Overall, the figure serves primarily as a visualization tool to help explore potential relationships between DDR markers and various tumor and patient-related features. This type of visualization is not intended for statistical testing and should be interpreted with caution."

Reviewer 3 Report

Comments and Suggestions for Authors

The submitted paper is very interesting and necessary, but some changes are required. Please provide all changes in the file with the track changes option, and in the rebuttal letter, please add the line number where the changes have been made.

1. Data, including in lines 94-122, should be put in the M&M section.

2. The introduction section has to end the study's aim.

3. M&M and results sections have to be divided into smaller sub-sections, with descriptive titles. Currently, after reading the M&M section, it seems that the authors only performed the IHC analysis, which is not true.

4. For each results section, in the material and methods section, the statistical analysis should be described in detail.

5. In the introduction, please higlihted the importance and innovations of this paper should be figured out in the introduction.

6. In Introduction part, the authors should clarify the limitations of current breast cancer treatments and how the study aims to address them

7. In the Discussion part, the authors should compare and contrast the findings with previous studies on the histaminergic system and breast cancer. Moreover, the potential avenues for future research based on the current findings should be discussed. This demonstrates the authors' critical thinking and helps readers evaluate the study's broader context.

8. Please consider citing doi: 10.3390/ijms25126546.

Author Response

Thank you very much for taking the time to review this manuscript. Please find the detailed responses below and the corresponding revisions in the re-submitted files.

Comment 1: Data, including in lines 94-122, should be put in the M&M section.

Response to comment 1: Thank you for your suggestion. We have moved the specified text from lines 94-122 in the introduction to the Materials and Methods section to ensure proper placement of the information. This information can be found now under a new subsection called "2.1. Study Design". This adjustment helps to maintain the focus of the introduction and provide a clearer description of the study's aims and background.

Comment 2: The introduction section has to end the study's aim.

Response to Comments 2: Thank you for your valuable feedback. We have revised the introduction to clearly state the aim of the study at the end of the section and to highlight the importance and innovations of our research. The revised introduction now concludes with the following statement: "This study aims to evaluate the expression of active DDR proteins and their impact on disease course and clinical outcomes in the adjuvant setting of breast cancer patients. By providing a detailed analysis of DDR marker expression, this research seeks to offer new insights into their potential as prognostic and predictive biomarkers, ultimately con-tributing to more personalized and effective treatment strategies for breast cancer."

Comment 3: M&M and results sections have to be divided into smaller sub-sections, with descriptive titles. Currently, after reading the M&M section, it seems that the authors only performed the IHC analysis, which is not true.

Response to comment 3: Thank you for your insightful comment. We have revised the Materials and Methods section to include more descriptive sub-sections, ensuring that each sub-section the results of the study is clearly delineated and the full scope of our analyses is appropriately represented. This adjustment helps to clarify that the study encompasses a range of methodologies beyond IHC analysis.

Comment 4: For each results section, in the material and methods section, the statistical analysis should be described in detail.

Response to comment 4: Thank you for your valuable feedback. We have revised the Materials and Methods section to include detailed descriptions of the statistical analyses corresponding to each results section. Additionally, we have added a subsection to explain the k-means and hierarchical clustering analysis performed to enhance visualization. This ensures that the statistical methodologies used for each analysis are clearly outlined and can be easily referenced.

Comment 5: In the introduction, please higlihted the importance and innovations of this paper should be figured out in the introduction.

Response to comment 5: Thank you for your valuable feedback. We have revised the introduction to clearly state the aim of the study at the end of the section and to highlight the importance and innovations of our research. The revised introduction, as previously stated answering to comment 2, now concludes with the following statement: "This study aims to evaluate the expression of active DDR proteins and their impact on disease course and clinical outcomes in the adjuvant setting of breast cancer patients. By providing a detailed analysis of DDR marker expression, this research seeks to offer new insights into their potential as prognostic and predictive biomarkers, ultimately con-tributing to more personalized and effective treatment strategies for breast cancer."

Comment 6: In Introduction part, the authors should clarify the limitations of current breast cancer treatments and how the study aims to address them.

Response to comment 6: Thank you for your insightful comment. We have revised the introduction to clarify the limitations of current breast cancer treatments and how our study aims to address them. The following sentences have been added to the introduction in lines 80-83 of the revised version of our manuscript: "Despite these advances, limitations in current treatments remain. These limitations include variability in treatment response and the potential for significant side effects, which underscore the need for more precise biomarkers to guide therapy decisions”.  Moreover, the following sentence has been added to the introduction in lines 97-101: “This study aims to evaluate the expression of active DDR proteins and their impact on disease course and clinical outcomes in the adjuvant setting of breast cancer patients. By providing a detailed analysis of DDR marker expression, this research seeks to offer new insights into their potential as prognostic and predictive biomarkers, ultimately contributing to more personalized and effective treatment strategies for breast cancer."

Comment 7: In the Discussion part, the authors should compare and contrast the findings with previous studies on the histaminergic system and breast cancer. Moreover, the potential avenues for future research based on the current findings should be discussed. This demonstrates the authors' critical thinking and helps readers evaluate the study's broader context.

Response to comments 7: Thank you for your valuable suggestion. We have expanded the discussion to include a comparison of our findings with studies on the histaminergic and dopaminergic systems in breast cancer. Specifically, we have discussed how increased histidine decarboxylase (HDC) gene expression, associated with better relapse-free and overall survival, highlights the prognostic value of the histaminergic system [doi.org/10.1038/s41416-019-0636-x]. Similarly, the role of dopamine receptors, particularly the upregulation of DRD2 and DRD3 and downregulation of DRD5, has been highlighted in recent studies as significant in breast cancer progression [doi:10.3390/ijms25126546]. These additions underscore the complex interplay between different signaling pathways in breast cancer biology. Two new citation (47, 48) have been added, accordingly to the text. Furthermore, we have discussed potential avenues for future research, emphasizing the need to explore intersections between these pathways to identify novel therapeutic targets.

Comment 8: Please consider citing doi: 10.3390/ijms25126546.

Response to comment 8: Thank you for your valuable suggestion. We have already addressed your comment, answering to comment 7. The citation has been added, improving the quality of our discussion.

Reviewer 4 Report

Comments and Suggestions for Authors

In this study "DNA DAMAGE RESPONSE IN EARLY BREAST CANCER: A PHASE III COHORT IN THE PHOBOS STUDY", the authors examine expression changes in DNA damage response proteins in patient that were treated with Epirubcin - Cyclophosphamide or Epirubcin - Cyclophosphamide and Docetaxel.

This manuscript is well written and easy to follow.

All figures are required and need.

The study is rather simplistic using IHC of different DDR markers in different tumor sample. The authors focused on kinases such as ATM and ATR to examine status of the DNA damage response in these tumor samples.

While simplistic this study does provide interesting results that could result in how breast cancer is treated. 

Author Response

Thank you very much for taking the time to review this manuscript. Please find the detailed responses below

Comment 1: In this study "DNA DAMAGE RESPONSE IN EARLY BREAST CANCER: A PHASE III COHORT IN THE PHOBOS STUDY", the authors examine expression changes in DNA damage response proteins in patient that were treated with Epirubcin - Cyclophosphamide or Epirubcin - Cyclophosphamide and Docetaxel.

This manuscript is well written and easy to follow.

All figures are required and need.

The study is rather simplistic using IHC of different DDR markers in different tumor sample. The authors focused on kinases such as ATM and ATR to examine status of the DNA damage response in these tumor samples.

While simplistic this study does provide interesting results that could result in how breast cancer is treated.

Response to comment 1: Thank you for your positive feedback on our manuscript and for appreciating the clarity and necessity of the figures presented. We acknowledge the reviewer's observation regarding the simplicity of our approach using IHC to examine DDR markers such as ATM and ATR in tumor samples. While we agree that our methodology is straightforward, we believe that it provides a robust and reliable measure of DDR activation, which is crucial for understanding the biological characterization, disease progression, and treatment responsiveness in breast cancer. Our study aimed to generate clinically relevant insights for early breast cancer patients, and we are pleased that the reviewer finds our results potentially impactful for breast cancer treatment. We believe that even with a simplistic approach, our findings contribute valuable information to the field and lay the groundwork for more detailed investigations into DDR pathways and their therapeutic implications. Thank you once again for your constructive comments and positive assessment of our work.

Round 2

Reviewer 2 Report

Comments and Suggestions for Authors

I have no further comments on the article. The authors made changes to the manuscript in accordance with the recommendations and comments of the reviewers.

Reviewer 3 Report

Comments and Suggestions for Authors

nonę